# A Comparative Study of Commit Representations for JIT Vulnerability Prediction

**Tamás Aladics** [1,2,*] **, Péter Hegedűs** [1,*] **and Rudolf Ferenc** [1]

1    Department of Software Engineering, University of Szeged, 6720 Szeged, Hungary; ferenc@inf.u-szeged.hu
2    FrontEndART Ltd., 6720 Szeged, Hungary
*    Correspondence: aladics@inf.u-szeged.hu (T.A.); hpeter@inf.u-szeged.hu (P.H.)

**Abstract:** With the evolution of software systems, their size and complexity are rising rapidly. Identifying vulnerabilities as early as possible is crucial for ensuring high software quality and security. Just-in-time (JIT) vulnerability prediction, which aims to find vulnerabilities at the time of commit, has increasingly become a focus of attention. In our work, we present a comparative study to provide insights into the current state of JIT vulnerability prediction by examining three candidate models: CC2Vec, DeepJIT, and Code Change Tree. These unique approaches aptly represent the various techniques used in the field, allowing us to offer a thorough description of the current limitations and strengths of JIT vulnerability prediction. Our focus was on the predictive power of the models, their usability in terms of false positive (FP) rates, and the granularity of the source code analysis they are capable of handling. For training and evaluation, we used two recently published datasets containing vulnerability-inducing commits: ProjectKB and Defectors. Our results highlight the trade-offs between predictive accuracy and operational flexibility and also provide guidance on the use of ML-based automation for developers, especially considering false positive rates in commit-based vulnerability prediction. These findings can serve as crucial insights for future research and practical applications in software security.

**Keywords:** commit representation; vulnerability prediction; just-in-time

## 1. Introduction

Software systems are becoming increasingly complex and interdependent, with thousands of lines of code being added and modified daily. As a result, software vulnerabilities are becoming more prevalent and pose a significant threat to the security and reliability of software systems. This is clearly apparent from cybersecurity firm Tenable's report of 2021 [1]. According to them, from 2016 to 2021, the number of reported CVEs increased at an average annual growth rate of 28.3%, accumulating to a 241% increase. Many of these are zero-day vulnerabilities that were disclosed in 2021 across a variety of popular software applications, leaving software vendors a short time to prevent exploitation.

In order to mitigate these vulnerabilities, it is crucial to identify and address them as early as possible during the software development process. A fitting time for identification is during code addition to the codebase in the version control system, i.e., at commit time. This process is usually referred to as just-in-time (JIT) vulnerability prediction [2]. Commits contain data on bug fixes, feature additions, code refactoring, and additional metadata in the form of commit messages and author information. By analyzing commits based on the contained data, software engineers can identify patterns that may indicate the presence of vulnerabilities [3].

Identifying and analyzing these vulnerable commits manually is a daunting task, especially in large software projects with thousands of commits. It is not feasible for human analysts to examine each commit, and the task may be error-prone due to the sheer volume of changes [4]. Manual efforts are also found to be lacking in terms of adaptiveness,

as vulnerabilities appear very rapidly in different forms and human resources are expensive. To address this challenge, researchers have developed machine-learning-based approaches to automatically identify and predict vulnerable commits [5].

One critical component of these approaches is the representation of commits. Commit representations capture the information contained in commits in a way that can be processed by machine learning algorithms, typically resulting in variable-length vectors. Representations are designed based on data contained in the commits such as the commit messages [6], the changed lines of code (patches) [7], or commit metadata and patches together [2,8]. Some approaches also use code metrics to supplement the commit metadata to predict potential vulnerability-contributing commits, such as VCCFinder [9].

Commit representations that use source code as input also vary in what form they take the source code and what granularity they parse it. Source code can be used in its raw form as raw text and processed by popular NLP methods, intermediate representation forms can be used such as the abstract syntax tree (AST) [7] or code metrics derived from the source code can also be used [9]. The granularity is also an essential factor, as it directly affects the usability of the representation and it varies a lot: the commit can be taken as a whole by aggregating the patches of all the changed files [8], or it can be trained at the file, class, method, or even line level. A finer granularity will contain fewer data but provide a more precise location of a suspected vulnerable piece of code than a coarser-grained one.

In this paper, the aim is to provide an overview of these factors of commit representations by comparing three different commit representations for vulnerability prediction: CC2Vec, DeepJIT, and a code-change-tree-based representation. As part of the research, an evaluation was performed of these representations in identifying and predicting vulnerable commits using two recently published datasets: a dataset based on ProjectKB, which contains vulnerability-introducing commits of Java open-source projects [10] and Defectors [11], a larger dataset that comprises a large number of defective pieces of Python source code with commit information. These findings can inform the development of more effective commit representation techniques for identifying and addressing vulnerabilities in software systems guided by the following research questions:

*RQ1*: How effective are commit representations in predicting vulnerability contributing commits?

*RQ2*: How does granularity affect the overall vulnerability prediction power?

Our work follows a typical structure of related work, methodology, experimental setup, results, conclusion, and future research. The related work (Section 2) provides an overview of the previous studies related to the topic of JIT vulnerability prediction and commit representation, Section 3 gives technical details about the candidate models, the methodology section (Section 4) describes the detailed approach used in the study, Section 5 presents the data and tools used, Section 6 presents the findings, a discussion (Section 7) interprets the results, Section 8 discusses the potential dangers to the validity of our approach, and the conclusion and future research (Section 9) summarize the study's results and suggest future directions for research.

## 2. Related Work

The identification and prediction of vulnerable commits have been studied extensively in recent years. In this section, we review the existing literature on commit representations for vulnerability prediction.

An earlier line of work entails using different attributes derived from the commit to reason about the code change's effects. Mockus et al. used predictors such as the size of lines of code added, deleted, and unmodified, measures of developer experience, the type of change, etc., to predict the quality of change in regard to inspection, testing, and delivery [12].

Kamei et al. uses source-code metrics and Kim et al. uses features extracted from the revision history of software projects [13,14]. Similarly, other works extract features from metrics and/or commit metadata to enhance the procedure of code quality assessment and

ultimately prevent vulnerabilities [9,15]. Usually, in these methods, the extracted features are fed into a machine learning model, such as an SVM or a random forest.

Even though metric-based approaches were an improvement over manual procedures, they still could not capture the semantic and syntactic relations between the code elements. Lomio et al. found that existing metrics were not sufficient enough and additional work was needed to find more appropriate ways to facilitate JIT vulnerability prediction [16]. A possible route for improvement was based on the insight that Hindle et al. provided in their work: source code is similar to texts written in natural languages, and as such, NLP methods can be leveraged to improve the related models' effectiveness [17].

Agreeing on this insight, many works in the field use the source code itself as input to find an improvement over metric-based approaches. Minh Le et al. proposed DeepCVA, a convolutional neural network that uses code changes (patches) and their contexts [18] for commit-level vulnerability assessment.

Hoang et al. describe DeepJIT, an end-to-end deep learning framework also based on the convolutional neural network architecture, which automatically extracts features from commit messages and code changes and uses them to identify defects [2]. In their follow-up work, Hoang et al. propose CC2Vec, a framework that also uses commit messages and code changes as input but uses a hierarchical attention neural network to produce generally usable vectors [8] They evaluated their method on log-message generation, bug-fixing patch identification, and just-in-time defect prediction and found that they outperformed previous works. Our study specifically focused on DeepJIT and CC2Vec due to their innovative use of recent advances of machine learning approaches, such as CNNs and HANs. These models, at the time of our investigation, were contending with each other as state-of-the-art approaches for JIT vulnerability prediction. Their emerging prominence in commit analysis suggested their strong potential, making them particularly relevant to our research objectives in the dynamic field of defect prediction.

Even though these methods are promising due to their relatively high predictive power, they take the source code as (preprocessed) text and they do not use the strictly structured nature of source code. The available intermediate representations, such as the abstract syntax tree (AST) and control flow graphs (CFG) are promising for providing more structural information, as shown in works related to source code representation and repair [19–22].

Change trees are specifically designed to keep the structural differences in changed codes [23]. They are constructed by using the ASTs of the pre- and postcommit states and discarding the AST paths that are the same in the two states. The resulting AST structure is called a change tree, as it only contains the relations that are changed as part of the code change. We selected this method as our third candidate due to its customizability. It can be tailored for any source code element with an associated AST, potentially capturing more nuanced information than standard feature extraction methods. This versatility makes change trees a valuable addition to our comparative analysis in exploring the landscape of vulnerability prediction.

In this paper, we compare three already mentioned commit representations for vulnerability prediction: CC2Vec, DeepJIT, and a change-tree-based model. Our evaluation is conducted using two datasets. The first is a dataset of vulnerability-introducing commits from Java open-source projects [10], which originates from a manually curated set of vulnerability-fixing commits known as Project-KB [24], developed by software company SAP (Walldorf, Germany). The second dataset, Defectors [11], encompasses source code files, with approximately 93,000 defective and 120,000 defect-free entries, drawn from 24 well-known Python projects.

## 3. Candidate Models

In this section, the focus is on the three machine learning models previously introduced for vulnerability prediction. We provide concise descriptions of each model, including their respective inputs and outputs. Additionally, this section aims to highlight the unique fea-

tures and differences among these models, offering insights into their individual strengths and applications in the context of vulnerability prediction.

*3.1. DeepJIT*

DeepJIT ([2]) is a machine learning model that is designed to analyze code changes by processing commit messages and corresponding code changes with the help of convolutional neural networks (CNNs). To enable feature learning, the model first encodes the raw textual data into arrays, which are then fed into the input layer. The input layer then embeds the commit message and code changes to create embedded representations.

DeepJIT employs two dedicated convolutional neural networks for processing commit messages and code changes. The commit message is fed into a convolutional layer with multiple filters to extract relevant features. Meanwhile, the code change is sent to a CNN that processes each added or removed code line using the first convolutional and pooling layers. These layers learn semantic features based on the words within the added or removed line. The subsequent convolutional and pooling layers aim to learn interactions between added or removed code lines with respect to the structure of the code change. The output of the CNNs consists of two vectors, one for the commit message and the other for the code changes. These vectors are aggregated using a feature fusion layer, to produce the final representation, which corresponds to the commit. In the following, a brief overview of the architecture follows; however, for a more in-depth understanding, please refer to the original paper.

As usual in NLP-related tasks, the input is first preprocessed with the help of the NLTK library [25], and each word in the commit message and code changes is represented as a fixed dimensional vector by using an embedding that is jointly learned with CNNs. Then, an independent process follows for the commit message and code changes for each commit, whose results are aggregated in the end.

For the commit message $m$, which is a sequence of words $w_1, \ldots w_{|m|}$, this process involves embedding the commit message into the matrix of embedded words $M \in \mathbb{R}^{|m| \times d_m}$, where $d_m$ is the embedding dimension. Then, to extract the commit message's salient features, a filter $f \in \mathbb{R}^{k \times d_m}$, followed by a ReLU activation function is applied: $c_i = ReLU(f * M_{i:i+k-1} + b_i)$, where $k$ is the window size, $b_i$ is the bias value, and * is a sum of element-wise product. The filter $f$ is applied to every k-words of the commit message and the outputs are concatenated to a vector $c = [c_1, \ldots, c_{|m|-k+1}]$. To characterize the commit message, as common in CNN networks, the filtering operation is followed by a pooling operation, which is in this case a max pooling operation over $c$, resulting in the embedded commit message $z_m$.

For the code changes in the commit, the input is more complex as there can be a variable number of corresponding files $[F_1, \ldots, F_n]$, where $n$ is the number of files in the code change. Each $F_i$ contains a number of lines (additions and removals), and each line has a sequence of words. Similarly to the commit messages, a word embedding is employed to obtain the embedded code changes' representation : $F_i \rightarrow \mathbf{F}_i \in \mathbb{R}^{N \times L \times d_c}$, where $N$ is the number of lines, $L$ presents the sequence of words in each line and $d_c$ is the word embedding dimension. For each line $N_i$, the same convolutional operations are performed as in the case of commit messages, to extract an embedding vector $z_{N_i}$. The embedding vectors for each line corresponding to the file $F_i$ are stacked to obtain a representation $\overline{\mathbf{F}}_i = [z_{N_1}, \ldots, z_{N_{|N|}}]$. Again, the convolutional and pooling layers are applied to extract the representation $z_{\overline{\mathbf{F}}_i}$ corresponding to the file $F_i$. As a result, for each change file $F_i$ $i \in C$, where C is the commit, we have the embedding vector $z_{\overline{\mathbf{F}}_i}$. For each i, the embedding vectors are concatenated to obtain a representation for the code change: $z_C = [z_{\overline{\mathbf{F}}_1}, \ldots, z_{\overline{\mathbf{F}}_n}]$.

With $z_m$ and $z_C$ calculated, they are concatenated and fed into a fully connected layer with a ReLU activation function. Finally, this vector is passed through a sigmoid function to provide a score that is trained to correspond to the vulnerability probability of the given commit C. The parameters are trained using the binary cross-entropy loss function with a regularization term, using the Adam optimizer [26].

*3.2. CC2Vec*

CC2Vec ([8]) is a deep learning architecture designed to learn vector representations of code changes in patches. The architecture can effectively embed code changes into a vector space, where similar changes are close to each other. The learning process is supervised using log messages written by developers to describe the semantics of the code changes. To be more precise, CC2Vec optimizes the vector representation of a code change to predict appropriate words, extracted from the first line of the log message.

To represent the code changes, CC2Vec analyzes scattered fragments of removed and added code across multiple files that are part of a commit. As related work suggests that using an attention mechanism can help to model structural relations ([27]), CC2Vec uses a specialized hierarchical attention network (HAN) to construct a vector representation of the removed and added code of each affected file in a given patch. The HAN first builds vector representations of lines, then constructs vector representations of hunks using these vectors, and finally aggregates these vectors to construct the embedding vector of the removed or added code. Then, CC2Vec employs multiple comparison functions that produce features representing the relationship between the removed and added code. These features are then concatenated to form an embedding vector for the affected file. Finally, the embedding vectors of all the affected files are concatenated to form the vector representation of the code change in a patch.

In more detail, in a commit there is a number of files ($F$); for each file, the added (removed) code is represented as a 3D matrix $B \in \mathbb{R}^{H \times L \times W}$ where $H$ is the number of hunks, $L$ is the number of added (removed) lines, and $W$ is the number of words. Since $F, H, L, W$ can differ significantly across different commits, each input is padded/truncated to the same length. For each dimension in $B$, a separate attention mechanism is employed to capture the structural information, referred to as feature extraction layers, which make up the main architecture of the HAN.

For instance, each word $w_{ijk}$ ($i \in 1 \ldots H, j \in 1 \ldots L, k \in 1 \ldots W$) is embedded into $\overline{w_{ijk}}$, a fixed-length vector using a learned word-embedding matrix $\overline{W} \in \mathbb{R}^{|V_C| \times d}$, where $V_C$ is the vocabulary, and $d$ is the embedding dimension. To capture the contextual information of the sequence of words (line), a bidirectional GRU is used, which includes embedding the line in a forward and backward directions. The forward GRU results in the embedded vector $\overrightarrow{h_{ijk}} = \overrightarrow{GRU}(\overline{w_{ijk}}), k \in 1 \ldots W$, and similarly, we obtain $\overleftarrow{h_{ijk}}$ for the backwards direction. The two vectors are concatenated to $h_{ijk}$ and fed to a fully connected layer with a ReLU activation function to obtain the hidden representation $u_{ijk}$ for the word $w_{ijk}$.

As of recent discoveries in the field of ML, sequence representation can be efficiently improved using the attention mechanism [28], which aims to learn the "importance" relations of the elements of a sequence. In the context of CC2Vec, at the word level, the authors employed "word attention", by first defining a word context vector $u_w$ (randomly initialized and learned during the training for each word), calculating the alignment scores $\alpha_{ijk} = softmax(u_{ijk}, u_w)$, to finally obtain a line vector for the line by calculating the weighted sum of the embedded vectors: $s_{ij} = \sum_k \alpha_{ijk} h_{ijk}$.

To remain concise, we do not show the rest of the architecture in detail, but similar mechanisms are used to get context-aware vectors for the line and hunk dimensions. After the input is ran through the HAN, the result is two vectors for the added and removed pieces of code $e_a$ and $e_r$, respectively. To aggregate these vectors, they are fed to the comparison layers, which employ a number of different comparison functions (such as element-wise subtraction, element-wise multiplication, cosine and Euclidean similarity, etc.). The output of the comparison layers are concatenated and passed through a final fully connected hidden layer. The parameters are learned using the Adam optimizer, employing cross-entropy with a regularization term as the loss function, conditioned to predict words in the commit message.

To summarize, CC2Vec uses the attention mechanism to model the hierarchical structure of a code change, a mechanism that helps to capture the structural dependencies in the code change, and thus, it can effectively learn vector representations of code changes.

### 3.3. Code Change Tree (CCT)

A code change tree ([23]) is a novel way of presenting source code changes by representing the differences between two states of source code at a structural level. To reason about the structure, an intermediate representation such as the abstract syntax tree (AST) is used. The localization of the prediction is performed at different levels such as statement, method, class, or file levels. Since the AST is used as the base structure, any source code element that has a corresponding AST can be represented by this method.

To represent only the changes, the authors designed a novel structure called the code change tree that captures only the differences between two ASTs. The CCT is constructed by first representing each tree as a set of unique paths from their root to each terminal. These paths are referred to as root paths, and are investigated in other works related to source code representation [29,30]. Then, the root paths that are identical in both trees are discarded from the reference AST's set of root paths. Finally, a tree is constructed from the reference AST's root paths, which represents the code change tree.

For a brief and not exhaustive but more formal definition of CCT, let $A = (N, E, r, \gamma)$ be an AST corresponding to a piece of source code, where $N = n_1, \ldots, n_m$ is the set of $m$ nodes, $r \in N$ is the root node, $\gamma$ is a mapping that maps the nodes to their children, and $E$ is the set of edges between the nodes, that is, if there is an edge between $n_i$ and $n_j$, then $(n_i, n_j) \in E$. Note that generally ASTs are more complex with more attributes, but they are omitted as they are not relevant for the definition of a CCT.

Any sequence of nodes $r = n_1, \ldots, n_k, k <= m$ is a root path, if $n_1 = r$, $|\gamma(n_k)| = 0$ and $(n_i, n_{i+1}) \in E$, that is, the path starts from the root node and ends in a terminal. By traversing the tree, we can obtain any number of root paths up to the number of terminals in the tree, so let $\alpha$ be the mapping that maps the tree to a set of root paths extracted from the tree. Also, let $A_{pre}$ and $A_{post}$ be the ASTs corresponding to the pre- and postchange states of a piece of source code. $C_{pre}$ is a code change tree representing the differences in $A_{pre}$ when considering $A_{post}$, if it is built by considering the subtractions of the root paths in $A_{post}$ from the root paths in $A_{pre}$: $C_{pre} = \alpha(A_{pre}) \setminus \alpha(A_{post})$. This way, any root paths found in $C_{pre}$ are unique in $A_{pre}$ and represent the structural (AST level) changes between $A_{pre}$ and $A_{post}$ with $A_{pre}$ being the reference point. Similarly, $C_{post}$ can be constructed to represent the structural changes between the two states, with the poststate being the reference point.

To adapt tree-based methods for use with machine learning (ML) models, a numeric transformation of the trees is required. A common way for this transformation is to traverse the tree and process the nodes in the traversal order. In our work, we traversed the change trees (both $C_{pre}$ and $C_{post}$) in a depth-first manner, padded or truncated them to the same length, and concatenated the two sequences. We also defined a vocabulary that held the different node types found in the trees and mapped them to unique integers using the Gensim framework [31], then mapped the nodes to their corresponding identifier (an integer value). For the next step, a way to transform the mapped node sequence to a meaningful representation is needed, which can be used for vulnerability prediction. In our work, the method of this transformation differed based on the dataset.

For the dataset derived from ProjectKB, we employed Doc2Vec to convert the sequences into fixed-length vectors [32]. This technique allowed us to generate document-level vectors that could be effectively utilized by a random forest model for classification tasks. The Doc2Vec model itself was trained on a corpus of 2 million Java methods, which were randomly sampled from the GitHub Java Corpus [33], and after training, it was used to embed the sequences into fixed-length vectors. In contrast, when handling the substantially larger Defectors dataset, the computational demands were increased for the use of random forest, and the whole dataset could not be loaded into memory, which was problematic, as most ML frameworks do not support iterative or batched training of random forest models. For these reasons, we implemented long short-term memory networks (LSTMs) to process the token sequences. We also tried one-dimensional CNN network on a smaller subset of the data, but they worked with a similar effectiveness, so

we decided to use LSTMs because of their simpler interface. After running through the LSTM layer, the resulting hidden representation was then passed through a simple dense layer for the vulnerability classification. During training, we used the binary cross-entropy loss function with the Adam optimizer to update the parameters.

As outlined in this technical discussion, code change trees offer an AST-like structure that keeps only paths that are unique to the chosen (pre- or postchange) source code state, that is, the parts of the AST that are changed with respect to the other state. This way, unlike previous methods, the downstream methods can benefit from the advantages of the AST structure for change representation: relations are clearly identifiable as edges, scopes can be explored trivially by traversing the tree, hidden or implicitly defined source-code elements are present (unlike in raw source code), etc.

## 4. Methodology

In this section, we outline the methodologies employed to analyze the three distinct models in our study. Given the fundamental differences in the architectures of these models, specific preprocessing was necessary to tailor the inputs appropriately. Below, we detail the unique processing requirements for each model.

*DeepJIT:* (https://github.com/soarsmu/DeepJIT, accessed on 2 January 2024) as their implementation does the necessary NLP preprocessing for their model (stemming, tokenization, removal of outliers in the vocabulary, etc.). However, we still needed to prepare the input as the implemented scripts expected it, which entailed that for each commit in the vulnerability datasets, we extracted the added and removed lines along with the commit messages and added the commit's corresponding label (introducing or nonintroducing).

*CC2Vec:* While CC2Vec is a model that is designed to output a commit representation that is generally usable, the authors in their implementation provided a modification of the model that was tailored for JIT vulnerability prediction (https://github.com/CC2Vec/CC2Vec, accessed on 2 January 2024). We used it with close to identical preparation to the case of DeepJIT.

*Code change tree:* For the code change tree (CCT) model, additional steps were necessary, differing between the two datasets. For the Project-KB based dataset, we opted for a method-level representation because the dataset's smaller size allowed for a finer granularity analysis without prohibitive computational costs. For each commit, we extracted every function that was altered and constructed their corresponding CCTs. As detailed in Section 4, these trees were then flattened and embedded using Doc2Vec to produce a vector for each method.

To ascertain the vulnerability of a method, we trained and evaluated a random forest model on the CCT vectors, as this was the best-performing traditional ML model reported by the authors. To determine commit-level vulnerability, we examined the predictions for every changed method within a commit; if any method was deemed vulnerable, the entire commit was labeled as vulnerable.

Regarding the Defectors dataset, its significantly larger size made the method-mining process computationally prohibitive. Furthermore, it was important to assess the applicability of CCT-based models at a higher level of granularity, a key issue under active investigation in our research questions. Consequently, we trained word embeddings for each token composing the flattened tree and inputted the sequence into a bidirectional LSTM layer, followed by a dense layer with a sigmoid activation function. This approach translated the LSTM's hidden state into a singular value indicative of the vulnerability probability.

## 5. Materials and Methods

The experimental setup was designed to evaluate the performance of the investigated methods and architectures. In this section, we give insight into the datasets, the prepro-

cessing tasks, and the metrics that were used for the evaluation in such an imbalanced environment: accuracy, F1-score, precision, and recall.

### 5.1. Datasets Setup

The first dataset utilized in our study was derived from Project-KB [24], a dataset curated by SAP, cataloging vulnerability entries each associated with a unique CVE identifier. Project-KB meticulously records the identified fixing commit for each vulnerability, along with supplementary metadata.

The methodology outlined in [10] details a process that transforms a database of vulnerability-fixing commits into one of vulnerability-introducing commits. Specifically, it employs a two-phase algorithm: initially, it identifies a pool of candidate vulnerability-contributing commits using the SZZ algorithm [34]. Given that SZZ has a tendency to produce numerous false positives and offers limited customization, the subsequent phase applies a targeted filter, utilizing relevance scores to assess the probability that a particular commit is the true introducer.

However, the dataset in its initial form contained insufficient entries for a substantive analysis, prompting us to manually augment it. Considering the extremely low likelihood that a randomly chosen commit would be vulnerability-inducing, we operated under the assumption that such commits were nonintroducing. Adopting this methodology, we paired each identified vulnerability-contributing commit with four noncontributing counterparts. It should be noted that on occasion, the process of randomly selecting commits was hindered by the unavailability of certain repositories, but generally, this sampling procedure proved to be successful.

As a consequence, the expanded Project-KB based dataset employed for our analysis comprised 474 positive instances (vulnerability-introducing commits) and 2114 negative instances (the corresponding fixing commits and the noncontributing commits sampled), culminating in a positive–negative ratio of 22.4% within the dataset.

Regarding the Defectors dataset, only minimal adjustments and refinements were necessary: we extracted the associated commit messages and reconstructed the precommit states for each record. Furthermore, any entries with empty "git diff" or "content" columns were excluded. It must also be noted that the authors of Defectors provided dedicated train and test splits, which we used. Consequently, the refined dataset used for training contained 91,177 vulnerable and 102,243 nonvulnerable files, resulting in a positive–negative ratio of 47.1%. For the testing split, it included 1246 vulnerable and 8754 nonvulnerable files, yielding a positive–negative ratio of 12.5%.

### 5.2. Datasets Structure and Example

Both datasets in our study have a similar structure, encompassing key metadata for vulnerability or defective commit analysis. This includes the commit SHA, repository identifier, filepath, and vulnerability labels. To give a clearer picture of how these datasets are structured, and to understand the nature of the defects and vulnerabilities our work aimed to examine, we present a straightforward example. This will help in illustrating the datasets' composition and inform our comparison of the candidate models.

In the Project-KB dataset, a typical entry, such as one involving the "Openfire" software in the "igniterealtime/Openfire" repository, is detailed with specific data like the commit hash "c9cd1e521673ef0cccb8795b78d3cbaefb8a576a" and the affected file "ConnectionManagerImpl.java". This example represents a fixing commit for a DOS attack vulnerability, labeled as "negative" (indicating it is not vulnerable). Project-KB also includes "positive" examples, where the commits introduced the vulnerability (in this case, the exposition to DOS attack), identified through a method briefly described later. Each entry in Project-KB correlates with a vulnerability listed in the National Vulnerability Database (NVD) (https://nvd.nist.gov/ accessed on 2 January 2024).

The Defectors dataset, while sharing common fields with Project-KB such as commit SHA and repository identifier, adds additional information like commit date and git diffs.

However, for our analysis, we primarily utilized the common fields. Also, Defectors encompasses a broader range of vulnerabilities and defects, not restricted to NVD entries but identified through manually designed patterns by its authors. To address potential false positives, they implemented multiple filtering methods.

Finally, it is important to note that the two datasets concentrate on different programming languages: Project-KB is compiled from open-source Java projects, whereas Defectors comprises Python projects. Consequently, our comparison not only serves to evaluate the methods but also to deduce the effectiveness of different approaches across programming languages.

*5.3. Metrics*

To assess the performance of the different algorithms and models on the dataset, we used multiple metrics. The most straightforward metric used to quantify a model's performance is accuracy, a metric that measures the overall correctness of a classification model. It is the ratio of the number of correct predictions made by the model to the total number of predictions made.

$$accuracy = \frac{TP + TN}{TP + TN + FP + FN} \tag{1}$$

However, in imbalanced datasets such as the one we presented in Section 5, where one class is significantly more prevalent than the other, accuracy may not be a good measure of model performance because the model may predict the majority class most of the time, resulting in high accuracy but poor performance on the minority class. To combat this, researchers usually employ metrics more descriptive of performance in an imbalanced environment.

One common metric is precision, which is a metric that measures the proportion of true positive predictions among all the positive predictions made by the model. This metric is important in situations where the cost of false positives is high.

$$precision = \frac{TP}{TP + FP} \tag{2}$$

Recall is another metric that complements precision in a way that it measures the cost of false negatives.

$$recall = \frac{TP}{FN + TP} \tag{3}$$

To describe the overall performance of a model with a single indicator, similar to accuracy but better suited for imbalanced datasets, the $F_\beta$ score can be used, which combines precision and recall with respect to the $\beta$ parameter. To be more precise, $F_\beta$ measures the effectiveness of retrieval with respect to a user who attaches $\beta$ times as much importance to recall as precision.

$$F_\beta = (1 + \beta^2) \cdot \frac{precision \cdot recall}{(\beta^2 \cdot precision) + recall} \tag{4}$$

In our work, we used $F_1$, $F_2$, and $F_{0.5}$ scores to reason about use cases where: recall is as important, twice as important, or half as important as precision.

**6. Results**

In this section, we present our findings and in the following section we interpret them in more detail by answering two research questions previously introduced, based on the results presented in Table 1. The results were calculated by training and evaluating the three methods investigated on the datasets as explained in Section 5.

**Table 1.** The performance measured by various metrics for each model, averaged over a tenfold cross-validation process for ProjectKB and evaluated on the test set for the Defectors dataset.

| Dataset | Model | Accuracy | $F_1$ | $F_{0.5}$ | $F_2$ | Precision | Recall |
|---------|-------|----------|-------|-----------|-------|-----------|--------|
| ProjectKB | DeepJIT | 0.74 | 0.47 | 0.41 | 0.56 | 0.38 | 0.64 |
| | CC2Vec | 0.59 | 0.37 | 0.3 | 0.51 | 0.32 | 0.64 |
| | CCT + RF | 0.7 | 0.33 | 0.3 | 0.37 | 0.29 | 0.4 |
| | Baseline | 0.65 | 0.22 | 0.22 | 0.22 | 0.22 | 0.22 |
| Defectors | DeepJIT | 0.71 | 0.35 | 0.27 | 0.47 | 0.24 | 0.63 |
| | CC2Vec | 0.75 | 0.39 | 0.31 | 0.50 | 0.28 | 0.63 |
| | CCT + LSTM | 0.66 | 0.3 | 0.23 | 0.42 | 0.20 | 0.58 |
| | Baseline | 0.78 | 0.13 | 0.13 | 0.13 | 0.13 | 0.13 |

In the case of the Project-KB based dataset, since the number of entries is limited, splitting the data into train–test splits would be highly influenced by randomness. To mitigate the effects of randomness, the values were calculated using a 10-fold stratified cross-validation process employing scikit-learn's implementation [35], and the averaged values are presented in Table 1. For more detailed information about the distribution of the scores, please refer to Figure 1, where the values corresponding to the accuracy, $F_1$, $F_2$, and $F_{0.5}$ metrics are illustrated in box plots. In the case of the Defectors dataset, which is sufficiently large, we used the train–test split sets that were provided with the dataset.

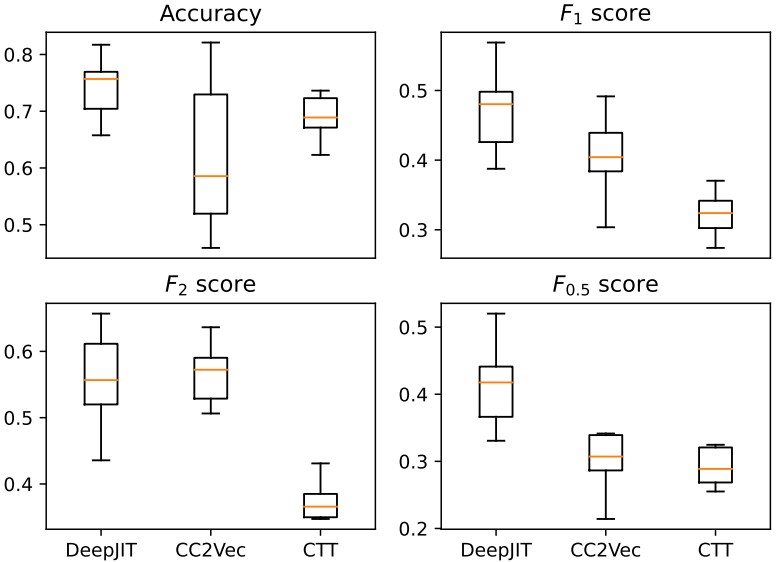

**Figure 1.** The distribution of accuracy, $F1$, $F_2$, and $F_{0.5}$ scores shown in box plots

To establish a simple baseline, we included metrics for a dummy classifier that labeled vulnerable entries according to the positive–negative ratio of the dataset (denoted as "Baseline" in Table 1). Specifically, this classifier predicted an entry to be vulnerable with a 22% probability for the Project-KB dataset and a 12.5% probability for the Defectors dataset, respectively.

## 7. Discussion

In this section, we delve into a comprehensive discussion on the state of just-in-time (JIT) vulnerability prediction. Our aim is to give insight into the strengths and disadvantages of the candidate methods by answering two research questions that stem from our experimental findings.

### 7.1. RQ: Can Commit Representations Be Used to Predict Vulnerability-Contributing Commits?

Based on our results, the investigated commit representation can be useful to a different degree depending on the use case. As a general verdict when the cost of false positives and

false negatives are equal, the $F_1$ score shows that all of the methods beat the baseline in both datasets.

Even though the improvement is noticeable, the baseline itself is quite simple, and we cannot conclude that these representations and architectures are sufficient enough by themselves for vulnerability prediction in their current state. However, they can still be used as supplementary tools for the manual workflow of vulnerability prevention and provide useful insights to security experts or notify suspicious commits for further inspection. As such, in the industrial environment, different use cases are possible when the cost of false negatives and false positives are not equal:

*The cost of false negatives is the priority:* Where the FNs are costly, a high recall value is desirable. In these cases, the $F_2$ score is a good indicator of the performance. In these cases, the investigated representation forms provide useful information as they beat the baseline by a larger margin. It is important to note that improving recall is fairly simple by being more permissive when labeling entries as vulnerable, which incurs a costly trade-off for precision.

*The cost of false positives is the priority:* In these cases, the $F_{0.5}$ score is a relevant metric to measure the effectiveness of the different approaches. Based on our results in these cases the commit representation methods are providing only marginal improvement over the baseline, which makes their usability questionable.

This use case is particularly important, as in many software development workflows, the cost of false positives is very high, and the human workforce will quickly abandon methods that make consistent errors, as shown in many empirical studies [36–38]. Because of this, for supplementary tools, high precision is fundamental so that they do not generate too much overhead work. As in this study, the investigated commit representations fall short in terms of precision and as such $F_{0.5}$ score, finding more appropriate methods that work with better precision (even at the cost of a lower recall) is an area where improvement would definitely be welcome, both by the industry and the research communities.

### 7.2. RQ: Can Commit Representations Be Used for More Localized Vulnerability Predictions?

The different representation forms investigated in our work differ not only in their architecture but in the granularity with which they can be used. DeepJIT and CC2Vec work at a commit level and are hardly customizable for finer-grained predictions as they use commit-level metadata. On the other hand, the CCT-based approaches can be used for any source code element that has a corresponding AST, making them arbitrarily customizable in terms of granularity.

In our work, we trained the CCT-based model for method-level predictions and aggregated the results as described in Section 5. Even though, as seen in Table 1, the overall performance of the CCT falls short compared to DeepJIT and CC2Vec, the noticeable difference comes from the recall, while in precision, it works similarly to the other approaches. As already discussed in the previous RQ, in many use cases in the software development process and industry, precision is the decisive factor because of the high cost of false positives. This finding, complemented by the finer granularity that CCT provides, makes it a good decision for situations where more localized vulnerability reports are expected.

### 8. Threats to Validity

In this section, we acknowledge several potential limitations and the measures taken to mitigate them. Concerns regarding the generalizability across different programming languages are partially addressed by incorporating datasets from both Java and Python. While this enhances the study's scope, it remains an area for future expansion.

The chosen evaluation metrics and their interpretation, particularly F-scores and accuracy, are heavily influenced by the distribution of the datasets. To mitigate the risks of misleading results from imbalanced datasets, we incorporated a dummy classifier as a

baseline. This approach helps illustrate the datasets' complexity and the relative predictive power of our models.

The number of investigated models (DeepJIT, CC2Vec, code change tree) was limited to three, which may not comprehensively cover the vulnerability prediction landscape. However, this selection provides valuable insights and serves as a foundation for future research to expand upon.

Different types of models were employed in the case of code change trees, notably the use of random forest on the Project-KB dataset and LSTM for Defectors. While our primary objective was to evaluate the effectiveness of commit representation forms rather than the underlying classifier models, to ensure consistency, we tested LSTM on the Project-KB dataset and found similar results to random forest, leading us to continue with the latter for alignment with previous research on code change trees.

The models demonstrate limited predictive power, making them not yet viable for practical, stand-alone use. To mitigate this, we provide a comprehensive discussion on potential use cases. This discussion highlights scenarios where prioritizing the minimization of either false positives or false negatives could be beneficial. Such prioritization is crucial in contexts where the cost of one type of error outweighs the other, allowing for more tailored and effective application of these models despite their limitations.

## 9. Conclusions and Future Research

In this study, we presented a comparative analysis of three different commit representations for vulnerability prediction, namely CC2Vec, DeepJIT, and code change trees (CCT), with the aim of providing insights into the landscape of just-in-time (JIT) vulnerability prediction. Our results showed that all three methods beat the baseline and that DeepJIT and CC2Vec were more effective for commit-level predictions, while CCT was more flexible and customizable for finer-grained predictions. Since at the time of this publication, CC2Vec and DeepJIT were state-of-the-art approaches for the topic of JIT vulnerability prediction (to the best of our knowledge), reasoning about their usage, performance, and comparison with the change-tree-based models can be interpreted as an analysis of the current state of JIT vulnerability prediction. To provide a more detailed discussion with real-world usage in focus, we identified different use cases where these representations could be useful for vulnerability prediction, depending on the cost of false positives and false negatives.

Our study has several implications for developers, security analysts, and researchers, as it provides valuable insights into the effectiveness of different commit representations for predicting vulnerable commits. However, there are several avenues for future research that can build on this work.

First, our study only investigated three commit representations, and there are several other representations that could be explored. For instance, future studies could investigate the effectiveness of graph neural networks, convolutional neural networks, or other types of tree-based models. Additionally, future research could also explore the combination of different representations to improve predictive performance.

Second, our study focused on vulnerability prediction at the commit level, and there is a need to investigate vulnerability prediction at different granularities. For instance, future research could investigate the effectiveness of commit representations for predicting vulnerabilities in specific source code elements such as methods, classes, or packages.

Third, our study used two datasets for training and testing, and future research could investigate the generalizability of the findings on other datasets. Additionally, future research could also investigate the effectiveness of the different commit representations on different programming languages and software systems.

Overall, our study provides valuable insights into the effectiveness of different commit representations for vulnerability prediction, and future research can build on this work to improve the security and quality of software systems.

**Author Contributions:** Conceptualization, T.A. and P.H.; methodology, T.A.; software, T.A.; validation, T.A., P.H., and R.F.; investigation, T.A.; resources, R.F.; data curation, T.A.; writing—original draft preparation, T.A.; writing—review and editing, T.A.; supervision, R.F. and P.H.; project administration, R.F. and P.H.; funding acquisition, R.F. All authors have read and agreed to the published version of the manuscript.

**Funding:** The research was supported by the the European Union project RRF-2.3.1-21-2022-00004 within the framework of the Artificial Intelligence National Laboratory and by project TKP2021-NVA-09, implemented with the support provided by the Ministry of Innovation and Technology of Hungary from the National Research, Development and Innovation Fund, financed under the TKP2021-NVA funding scheme. The work was partly supported by the EU-funded project Sec4AI4Sec (grant no. 101120393) as well.

**Data Availability Statement:** The data we used for evaluation can be found at the following links: Project-KB based dataset: https://zenodo.org/records/5855085 (accessed on 12 November 2023), Defectors dataset: https://zenodo.org/records/7708984 (accessed on 12 November 2023).

**Acknowledgments:** The authors would like to thank Árpád Beszédes for his invaluable contributions to this work.

**Conflicts of Interest:** The author Tamás Aladics was employed by the company FrontEndArt Ltd. The remaining authors declare that the research was conducted in the absence of any commercial or financial relationships that could be construed as a potential conflict of interest.

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
