# Peer review of "A Comparative Study of Commit Representations for JIT Vulnerability Prediction"

_computers, doi:10.3390/computers13010022_

Round 1

Reviewer 1 Report

Comments and Suggestions for Authors

The introduction is informative and descriptive of the problem being addressed and the solution proposed. Clear and concise research questions outlined. Adequate literature review.

Section 3 is limited to a brief, superficial and exempt of mathematical description of the vulnerability prediction models. The description included in the last paragraph in section 3 requires further technical details, along with appropriate description of the rationale that drives the approaches followed.

Section 4 could have presented samples of the two datasets used in this work. The description of the evaluation metrics does not describe the confusion matrix that allow computing the different metrics listed.

This work lacks discussion or explanation of the vulnerabilities being detected. Examples of the vulnerabilities within the datasets are required.

Comments on the Quality of English Language

Adequate use of English

Author Response

Thank you for your review! We valued your insights and tried to improve our paper appropriately, in line with your remarks. We attached the modified version of the manuscript.

  •  "Section 3 is limited to a brief, superficial and exempt of mathematical description of the vulnerability prediction models. The description included in the last paragraph in section 3 requires further technical details, along with appropriate description of the rationale that drives the approaches followed.""
    • In line with other reviewers remarks, we moved the models' description to a separate section from Methdology (Section 3 - Candidate models). Here, we provide a more detailed description of the candidate approaches, with technical details, formal definitions and their brief interpretations. We try to emphasize the methods relevance as our choice of candidate models and how their architecture justifies that.
  • Section 4 could have presented samples of the two datasets used in this work. The description of the evaluation metrics does not describe the confusion matrix that allow computing the different metrics listed.
    • We created a new subsection under "Section 5. Material and Methods", namely "5.2. Datasets structure and example", where we detail more in-depth the vulnerability datasets' structure, and give an example from one of the corresponding dataset's entry. Since the other dataset has very similar setup, with the only exception that it's of python source codes, we included only one example.
    • Unfortunately, due to the limited time available for revision, we were unable to include confusion matrices in our analysis. At the time of the initial publication, we had only stored the calculated F-scores, accuracy, recall, and precision metrics. Re-running the models on both datasets to generate these matrices would be impractical given the time constraints.
  • This work lacks discussion or explanation of the vulnerabilities being detected. Examples of the vulnerabilities within the datasets are required.
    • Connecting to our previous answer, in Section 5.2. "Datasets structure and example" while showing an example of the a dataset entry, we also try to address this concern by providing information about the vulnerability itself(not only the metadata stored immediately in the dataset).

Reviewer 2 Report

Comments and Suggestions for Authors

+ The presented research methodology and comparative analysis are correct. 

+ The language and editing of the paper are appropriate.  

+ The paper reviews the related works well, and the authors reference publications mainly from the last 5 years concerning methods of code vulnerability prediction.   

- The paper would have benefited from a justification for choosing the three methods mentioned and not others.

Author Response

Thank you for your review! We tried to improve on our paper according to your opinion:

  • The paper would have benefited from a justification for choosing the three methods mentioned and not others.
    • We have added some additional justification for our choices in "Section 2. Related Work", extending the corresponding paragraphs for DeepJIT & CC2Vec, and Code Change Trees. We have also added a new "Section 3. Candidate models", where we explain the models more formally and give more insight in their unique approaches and why are they selected as our chosen methods.

We have attached the revised manuscript.

Reviewer 3 Report

Comments and Suggestions for Authors

The paper provides a comparative study of three methodologies for JIT vulnerability prediction. Two of the selected methodologies represent the existing state of the art and act as the benchmark while the third one is the result of the authors' work. The paper examines an interesting topic that is worth further exploration. However, I feel that its main weakness is that key aspects of the domain and presentation are treated lightly and glossed over. For instance, authors should detail the criteria (inclusion/exclusion) that resulted in the inclusion of the proposed methodologies for evaluation. The related work section must be significantly improved in order to reflect why the selected methodologies have emerged as class-leading. Authors do not make it clear why AI-driven methods are superior to manual methods backed by developer experience (with the exception of execution speed). I don't think the dummy, baseline classifier brings any contribution to the results and could probably be omitted. Further more, the CCT methodology could be described in more detail, so that it can be fully understood without referencing other works. The threats to validity section, which is important in all case studies is missing. Some more minor observations:

  • "Departmet" misspelling
  • Commit Representations should be detailed earlier than in the Methodology; the methodology section should be dedicated to describing the case study and how authors propose to use it in order to provide answers for the proposed RQs.

Comments on the Quality of English Language

-

Author Response

Thank you for your insightful review. We greatly appreciate your feedback and have revised our paper accordingly:

  • However, I feel that its main weakness is that key aspects of the domain and presentation are treated lightly and glossed over. For instance, authors should detail the criteria (inclusion/exclusion) that resulted in the inclusion of the proposed methodologies for evaluation. The related work section must be significantly improved in order to reflect why the selected methodologies have emerged as class-leading.
    • We have improved on "Section 2. Related Work" to give more justification of the selected methods CC2Vec, DeepJIT and Code Chane Tree. We extended the corresponding paragraphs.
  • Authors do not make it clear why AI-driven methods are superior to manual methods backed by developer experience (with the exception of execution speed).
    • We believe that we touch upon this issue elaborately in the Introduction, apart from executing speed as already mention, we reason about the issues regarding human adaptability and development speed in contrast to the rate of vulnerability emergence (paragraph 1) and robustness software projects size (paragraph 3). Also we do not explicitly state that AI-based solutions are superior the humans, rather we discuss it in "Section 7. Discussion, in RQ 7.1". Here when discussing "Cost of false positives is the priority:" (which is the case most of the time), we interpret our results to show that AI-based approaches are currently more useful as supplementary tools to humans.
  • I don't think the dummy, baseline classifier brings any contribution to the results and could probably be omitted.
    • The dummy classifier serves the purpose of showing a simple baseline, that is beneficial to show the naive approach in the different environments where straightforward comparison is difficult. Since the different datasets might have different distributions and may be imbalanced to a high degree, a simple random guesser could have very good or bad results based on the dataset. With our dummy classifier we aimed to give a simple baseline to show that how "easy" is the dataset to predict. We touch this concern in the newly added "Threats to validity" section, see below.
  • Further more, the CCT methodology could be described in more detail, so that it can be fully understood without referencing other works.
    • We dedicated a separate section, "Section 3. Candidate models" (and moved Methdology to a Section 4. as requested) to give more insight to all of the models architecture. We provide a more formal definition of the models with brief interpretations, and towards the end some justification about their performance with relation to their architecture. This includes CCT methdology.
  • The threats to validity section, which is important in all case studies is missing.
    • We have added the new, "Section 8. Threats to validity" where we mention several issues that we found to be problematic, and if applicable, the measures we tried to take.
  • Commit Representations should be detailed earlier than in the Methodology; the methodology section should be dedicated to describing the case study and how authors propose to use it in order to provide answers for the proposed RQs.
    • We have moved the model descriptions into a separate section (Section 3.), while kept Methdology separate (Section 4.), short and focusing on the models preparation for the case study.

Please find the updated manuscript attached.

Round 2

Reviewer 1 Report

Comments and Suggestions for Authors

Reviewer's comments have been addressed.

Comments on the Quality of English Language

Adequate use of English.

Reviewer 3 Report

Comments and Suggestions for Authors

The paper was improved since version 1 and I believe it is ready for publication. There is 1 typo on line 238, with "show show"